# Telemedicine via data glasses in CBRN protection suit—Evaluation of medical qualification and technical feasibility

Sarah Bovenkerk[1]*, Anna Mueller[1], Rolf Rossaint[1,2], Michael Czaplik[1,2], Andreas Follmann[1]

1 Departement of Anaesthesiology, Faculty of Medicine, RWTH Aachen University, Aachen, Central Europe, Germany, 2 Docs in Clouds Telecare GmbH, Aachen, Central Europe, Germany

* acih@ukaachen.de

## Abstract

### Objective

Telemedicine in the context of chemical, biological, radiological and nuclear threats (CBRN) must adapt to the special features of the CBRN protection suit (hazmat suit). In a simulation, telemedicine with data glasses (smart glasses) was examined for its technical feasibility and the minimum required medical qualification.

### Methodology

The study was designed as an intervention study. A medical scenario was developed in which paramedics with four different medical qualifications were to provide initial care to a contaminated patient. Using data glasses worn in the CBRN protection suit, a telemedicine physician directed the simulation via video streaming. The times and attempts for each measure were examined, as well as the users' evaluation of two different data glasses.

### Results

A total of 40 participants were enrolled in the study. There were no significant differences could be found between participants of the next higher qualifications in terms of the duration of the guided measures and the success. Significant differences only occurred when comparing all evaluated qualifications to each other. Both data glasses have difficulties in the CBRN protection suit. The participants were not satisfied with the wearing comfort and the technical limitations of the glasses.

### Summary

In conclusion, telemedicine is feasible for emergency responders regardless of qualification in CBRN operations, although the data glasses currently appear unsuitable. Alternative hardware should be used and evaluated.

**Data availability statement:** All data are available under DOI: 10.6084/m9.figshare.26380267. The script is provided as supplement material.

**Funding:** Funded by Federal Office of Civil Protection and Disaster Assistance during the TeleSAN project (FKZ: 41201/45). The funders had no role in study design, data collection and analysis, decision to publish, or preparation of the manuscript.

**Competing interests:** MC (CEO, co-founder) and RR (co-founder) are with Docs in Clouds TeleCare GmbH, a company developing tele-medicine software. The other authors declare no conflicts of interest. This does not alter our adherence to PLOS ONE policies on sharing data and materials.

## Introduction

The nuclear disaster in Fukushima in 2011, the SARS-CoV-2 pandemic and chemical accidents in industrial plants are examples of chemical, biological, radiological and nuclear (CBRN) hazards. In this context, a disaster is defined as an existing imbalance between the need for assistance and the ability to provide it [1].

In CBRN situations, civil protection forces are responsible for decontamination and initial medical care of patients. Ideally, patient decontamination should occur prior to medical treatment. However, life-saving immediate measures such as securing the airway and stopping bleeding are exceptions to this and are urgently required in life-threatening conditions to prevent death. However, simply donning the necessary protective clothing requires some time. Additionally, it takes at least 30 minutes to set up a decontamination unit [2]. Therefore, it is even more important that life-saving measures are mastered quickly and safely. Another challenge in these situations is the large number of casualties that need to be rapidly assessed and treated [1].

It can be assumed that there will be an increased influx of patients at decontamination facilities. Until their decontamination, the clinical condition of an initially stable patient can deteriorate rapidly. Possible scenarios could include buried patients with severe soft tissue or amputation injuries, where life-threatening bleeding occurs due to loss of external compression after technical rescue. Measures may be necessary that go beyond life-saving immediate actions. However, the medical assistance is limited to the training level of the available helpers.

Like in many countries, civil protection in Germany relies to a large extent on the principle of volunteer helpers [3]. The concept of civil protection includes different levels of training (Table 1), which are roughly equivalent to the medical qualifications in the English-speaking countries with slight variations [4]. The lowest level of training focuses on life-saving immediate actions and is equivalent to the Emergency Medical Responder (EMR). Building on that is the qualification that can be compared to the Emergency Medical Technician-Basic (EMT-B). EMT-Bs are familiar with the basic equipment of an ambulance, and their training provides knowledge that goes beyond life-saving measures. The ranking qualification, like the Emergency Medical Technician-Intermediate (EMT-I) represents the highest non-medical training level in civil protection. In a three-month training the EMT-I learns theory and practice related to the most important medical measures and conditions, qualifying them not only for civil protection but also as part of an emergency service crew. The highest-ranking non-medical qualification is comparable to the Emergency Medical Technician-Paramedic (EMT-P). In a three-year education, the EMT-P learns all relevant measures to treat patients in civil protection and emergency medical services. However, due to a physician-supported rescue system in Germany, certain tasks such as application of specific medications like opioids as fentanyl or adenosine for treatment of supraventricular tachycardia are reserved for physicians, unlike in other countries.

Currently, EMT-P in Germany are only allowed to administer certain medications without consulting a physician. This regulation applies to clearly defined medical conditions and allows the administration of life-saving medication in medical emergencies, such as the intravenous administration of adrenaline. However, the

**Table 1. Medical qualifications in German civil protection.**

| Medical qualification | | Scope of the theoretical-practical training | Traineeship emergency service and hospital |
|---|---|---|---|
| German term | English term | | |
| **Sanitätshelfer** | Emergency Medical Responder (EMR) | 60 h | – |
| **Rettungshelfer** | Emergency Medical Technician-Basic (EMT-B) | 80 h | 80 h |
| **Rettungssanitäter** | Emergency Medical Technician-Intermediate (EMT-I) | 360 h | 160 h |
| **Notfallsanitäter** | Emergency Medical Technician-Paramedic (EMT-P) | 1920 h | 2680 h |

administration of medication is reserved for EMT-P, with the exception of the administration of oxygen and an electrolyte solution, which can also be administered by other qualified non-medical emergency personnel.

However, a problem lies in the fact that highly qualified helpers, physician and EMT-P, are rarely found in civil protection, as they are usually employed full-time in emergency medical services or hospitals or are needed in other roles. The physicians designated for civil protection would have to be recruited from hospitals or medical practices in a real disaster situation, where an increased patient load can also be expected there. At the same time, there is already a shortage of physicians today, which is projected to worsen in the coming years [5].

One solution could be the utilization of telemedicine. It has been used in prehospital emergency medical services for many years [6]. Through appropriate technical equipment, real-time vital data transmission as well as audiovisual communication with a telemedicine physician are possible The telemedicine physician can delegate medical measures, such as medication application, to the on-site personnel, which, due to legal restrictions, would not be allowed without a physician's delegation [7].

Apart from isolated deployments during natural disasters such as earthquakes or hurricanes [8], telemedicine has hardly been used in civil protection [9,10].

One study about telemedicine in civil protection shows, that physician delegated medicine based on paramedics like EMT-I is feasible [11], but without testing an CBRN scenario. To date, there is a lack of data regarding the use of telemedicine systems in CBRN incidents, as well as the testing of telemedical delegation of medical interventions to non-physicians with various qualifications. To address these aspects, our study simulates a CBRN scenario involving different medical measures, where data glasses were utilized for telemedicine. Data glasses have already been sporadically tested in prehospital settings [12,13], and were able to convince users above all with their hands-free operation with integrated voice control [14,15].

The aim of this study was to examine the technical feasibility of telemedicine in CBRN protection in addition to test which minimum medical qualification is necessary. The study shows that telemedical guidance of all medical qualifications is possible. However, the chosen technology in the form of data glasses is not considered suitable.

## Methods

### Study design

This study was designed as an intervention study. For this purpose, a medical scenario was developed in which the participants, acting as emergency responders, were required to provide initial care to a patient (represented by a Trauma Trainer Professional, Ambu GmbH, Bad Nauheim, Germany) in a CBRN situation under telemedical guidance using data glasses. Four groups with qualifications ranging from EMR to EMT-P were formed. A control group could not be used as the legal basis does not allow the responders to carry out some of the measures without physician delegation.

In addition to the telemedicine expert's visual monitoring of the individual measures, an on-site observer recorded various parameters for statistical analysis in a standardized protocol. The study also involved the participants completing

a questionnaire before and after the simulation. Questionnaire 1 before the study asks about the status quo: age, qualification, whether the participant has already worked with telemedicine or wearing a CBRN protection suit. Questionnaire 2 after the study with questions on the evaluation of the data glasses, the telemedicine instructions and a self-assessment of the implementation of the measures. A written declaration of consent was obtained from each study participant before the practical study. The practical implementation of the study took place between May 18, 2022 and March 3, 2023, after approval by the Ethics Committee (EK 052/22) of the University Hospital RWTH Aachen.

### CBRN protection suit

In the study, the participants wore, in addition to liquid-proof gloves (Alphatec 58-270, Ansell, Brussels, Belgium) and rubber boots, the powered air-purifying chemical protection suit Chemical Grey Inside (PM Atemschutz GmbH, Mönchengladbach, Germany) with an integrated respirator hood. This suit featured a continuous supply of fresh air through an internal blower, without causing an increase in breathing resistance, eliminating the need for prior medical examinations of the participants.

### Study participants

A total of 40 participants with different medical qualifications in civil protection were included in this study. The participants were evenly distributed across the training levels of EMR, EMT-B, EMT-I, and EMT-P. They were recruited through local relief organizations. The participants needed to have one of these qualifications without being in training for a higher qualification. The participants also had to be of legal age. Pregnant women were excluded from the study. The telemedicine physician, who was educated as an emergency physician, accompanied all study sessions.

### Telemedicine

The first 20 participants used the M300 data glasses model (Vuzix, West Henrietta, New York) for teleconsultation. Regardless of their medical qualifications, after 20 participants, the data glasses model was switched to the HMT-1 (Realwear, Vancouver, Canada). Due to an irreparable technical defect that affected not only the image transmission but also the teleconsultation, the data glasses had to be replaced. Both devices utilized an app based on a certified video consultation software (Docs in Clouds TeleCare GmbH, Aachen, Germany). The teleconsultation with the telemedicine physician was initiated through voice control, allowing the physician to observe the simulation in real-time via the integrated camera and video streaming. No video feed of the telemedicine physician was transmitted to the responders. Both data glasses gave real-time audio transmission to the telemedicine physician. Communication was facilitated through the built-in speakers and microphones of the data glasses. As the Trauma Trainer Professional was unable to generate real vital signs, simulated vital parameters were displayed to the participants on an external screen.

### Simulation scenario

The thematic background was an explosion at a chemical company resulting in the release of irritant gases. The selection of medical measures was deliberately based on typical injury patterns of a patient in CBRN situations. The responders encountered a 35-year-old patient (simulator) who was approximately 1.75 m tall and weighed around 75 kg. The patient had a left thigh amputation with a spurting bleed. The observer simulated screaming and also conducted the simulated patient communication. Due to the patient not being decontaminated yet, wearing a filter-powered air-purifying suit was necessary.

The patient needed initial care, following the x-ABCDE approach, which is commonly used in emergency medicine. In addition to life-saving immediate measures, spot decontamination of the affected body region was also required for further medical interventions. The telemedicine physician supported the participants by delegating and explaining different measures (Fig 1). To avoid statistical bias, the telemedicine physician strictly adhered to a predetermined script (Supplement

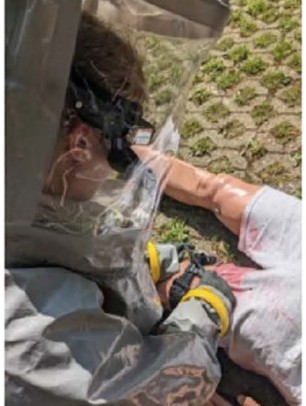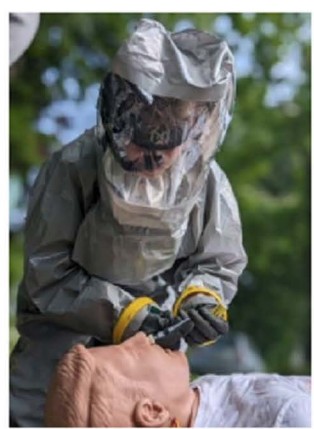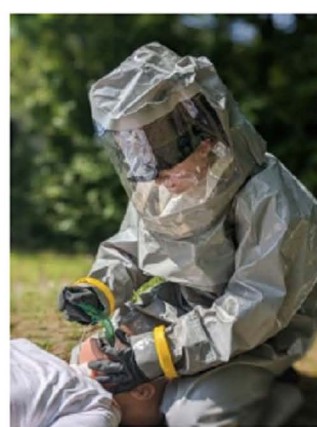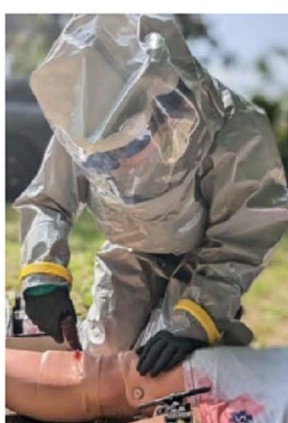

**Fig 1. Medical care of a contaminated patient in CBRN protection tourniquet, nasal application of medication, insertion of laryngeal mask, and intraosseous access.**

Material), which guided the study participant chronologically through the scenario and contained instructions for all medical measures. The script was written to guide all levels of qualification. All participants were therefore read the instructions in full, even if they did not necessarily need them to carry out the measure. The measure was also allowed to be carried out during the instructions.

After applying a tourniquet, the simulated vital signs, monitored by a telemedicine device showing mainly oxygen saturation rate and blood pressure to the participant and the telemedicine physician initially remained stable, and the patient continued to respond to verbal stimuli indicating severe pain, leading to the nasal application of 25 mg of esketamine. Shortly after, the patient experienced respiratory insufficiency and decreased consciousness, prompting the telemedical-guided insertion of a laryngeal mask and ventilation using a bag-valve-mask, significantly improving oxygenation. For further application of medication, an intraosseous (I.O.) access was established in the proximal tibia of the right lower leg with an Arrow EZ-IO Intraosseous Vascular Access System (Teleflex Inc., Wayne, USA). The medical scenario concluded with the initiation of crystalloid fluid infusion through the I.O. access. Other measures were explicitly excluded from the scenario. In addition, blood pressure measurement and oxygen saturation testing were carried out by the participants, they were excluded from the study examinations, as all participants were qualified to take the vital signs independently.

## Study procedure

After the participants had been informed of the content and purpose of the study, they were instructed on donning the CBRN protection suit, the data glasses and the available emergency medical equipment. The first questionnaire was filled out. The data glasses were started using voice commands before dressing up. Once fully dressed and briefed on the operational situation, the scenario began. The arrival of the responders at the simulator marked the starting point for documentation. The participant went through the simulation as described in paragraph 2.5, being guided and supported during the interventions by the telemedicine physician following a predetermined script. The observer accompanied the participant on-site without actively interfering in the process, documenting the timing, attempts and success of each intervention for later statistical analysis. The practical simulation ended with the conclusion of the medical scenario. Finally, the second questionnaire was completed.

## Target parameters

To assess the minimum qualification, the required time for the performed interventions was evaluated as the primary parameter. Additionally, secondary parameters included documenting the number of practical attempts needed to

successfully perform each intervention and assessing success or failure. Failure was recorded if either the allowed number of attempts was exceeded or if the intervention was deemed unsuccessful after evaluation by the observer or telemedicine physician at the end of the scenario. In a subgroup analysis, the influence of professional experience on the participants' performance was evaluated.

The moment of retrieving the necessary materials from the emergency backpack for executing each individual intervention defined the starting point for time measurement. Measurement ended upon successful execution or after exceeding the allowed number of attempts. Two attempts were allowed for the tourniquet, nasal medication administration, and intraosseous access, while three attempts were allowed for laryngeal mask placement, similar to what is postulated in difficult airway guidelines [16].

For the tourniquet, detachment and repositioning were considered separate attempts. Incomplete lockings, which were checked afterward, were deemed failures. Correctly locking the tourniquet in its designated holder was crucial. Medication administration was considered successful if the dosage and method of administration were performed correctly. Drawing up the medication again or repositioning laryngeal mask were considered an additional attempt. To protect the simulation mannequin from damage, the on-site observer corrected the needle position if the misplacement was not apparent to the telemedicine physician. This was considered an additional attempt. If the correction of the needle position was instructed by the telemedicine physician before drilling, this did not count as an additional attempt. The position of the intraosseous access was rechecked afterward.

Technical feasibility was assessed based on a stable connection. This involved counting the number of practical attempts required to establish a teleconsultation, as well as the number of connection interruptions during the scenario for both sets of data glasses. Additionally, requests from the telemedicine physician for head movement to adjust the camera position and viewing angle were counted.

Using a four-point Likert scale, the participants evaluated the user-friendliness of the data glasses in a second questionnaire. The following statements were assessed: "The data glasses could be worn safely and comfortably with CBRN protection suit," "Wearing the data glasses impaired my field of view," "The voice connection to the telemedicine physician was always consistently problem-free," and "The volume and sound quality of the connection were consistently good."

## Statistical analysis

IBM SPSS Version 26 (IBM Corp., Armonk, NY, USA) was used for performing statistical analyses. Significant group differences were calculated using the Kruskal-Wallis test. Kendall's $\tau$ and the Spearman rank correlation coefficient were employed for correlation calculations. The significance level was set at $p \leq 0.05$.

## Results

A total of 27 men (67.5%) and 13 women (32.5%) participated in this study. The age ranged from 19 to 44 years (mean age of 28.08 years). All 40 participants were included in the study analysis. One participant experienced a premature termination of the teleconsultation due to a technical failure of the data glasses, resulting in the practical simulation being terminated just before the establishment of the I.O. access. This participant was excluded from the analysis for the task of I.O. access. For comparability, participants were divided into subgroups based on their qualifications, as shown in Table 2.

### Time

For the primary outcome parameter of time, no statistically significant difference could be found between immediately consecutive medical qualifications, such as EMT-P and EMT-I, for any individual task. However, pairwise comparisons revealed significant differences (p = 0.17), as shown in Figs 2–4.

 

**Table 2. Classification of study participants into subgroups according to their medical qualification. In addition to the number of participants per group, gender distribution, and age, the distribution of the data glasses model (Vuzix M300, Realwear HMT-1) within the group is also listed.**

| Qualification | Quantity | Gender | | Age (years) | | Data glasses model | |
|---|---|---|---|---|---|---|---|
| | | women | men | average | range | M300 | HMT-1 |
| EMR | 10 | 6 | 4 | 27.1 (± 7.156) | 18-39 | 9 | 1 |
| EMT-B | 10 | 5 | 5 | 27.7 (± 7.484) | 19-40 | 4 | 6 |
| EMT-I | 10 | 1 | 9 | 26.3 (± 5.122) | 20-37 | 6 | 4 |
| EMT-P | 10 | 1 | 9 | 31.2 (± 7.068) | 20-44 | 1 | 9 |

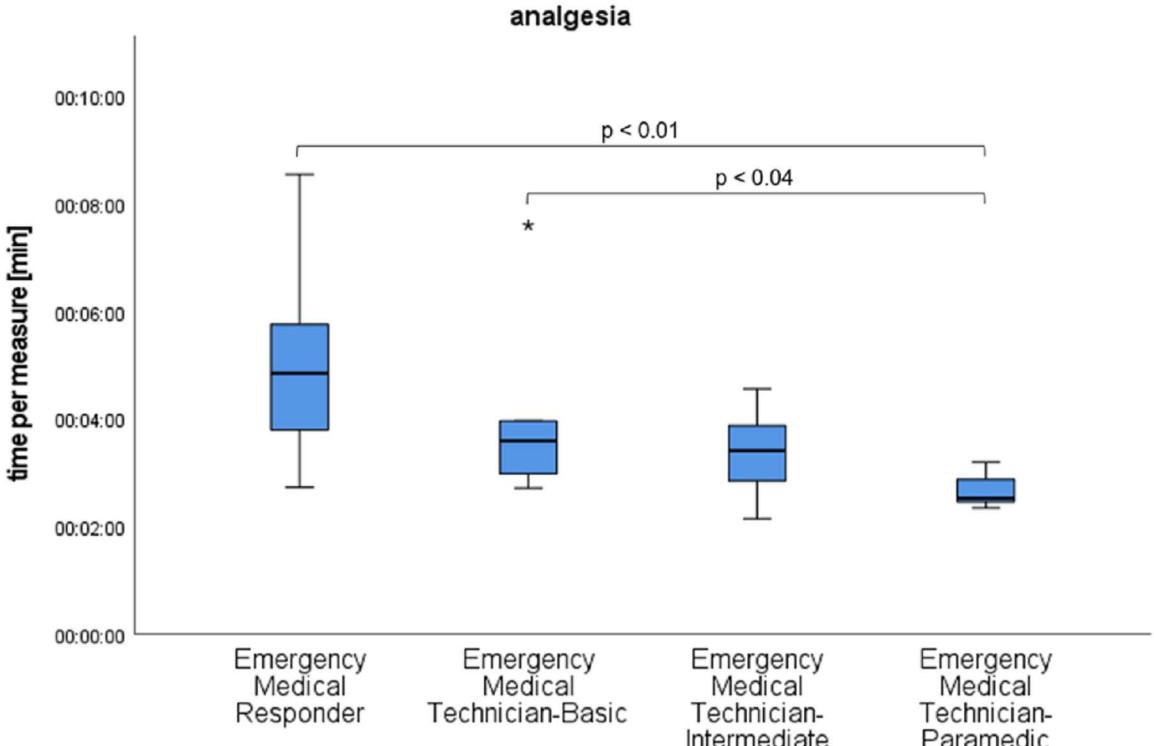

**Fig 2. Time results of the qualification subgroups for nasal medication application with significant differences marked.**

## Attempts

For the number of attempts, no statistically significant difference could be observed between the subgroups for any individual medical intervention, including non-skipping qualifications.

## Success

The success of performing a medical intervention varied both depending on the intervention itself and the qualifications of the helpers. The insertion of a laryngeal mask was successfully performed by all participants regardless of their qualifications. The tourniquet was correctly applied by all participants with the qualification of EMT-I, while one participant in each of the three other groups performed the securing inadequately tight.

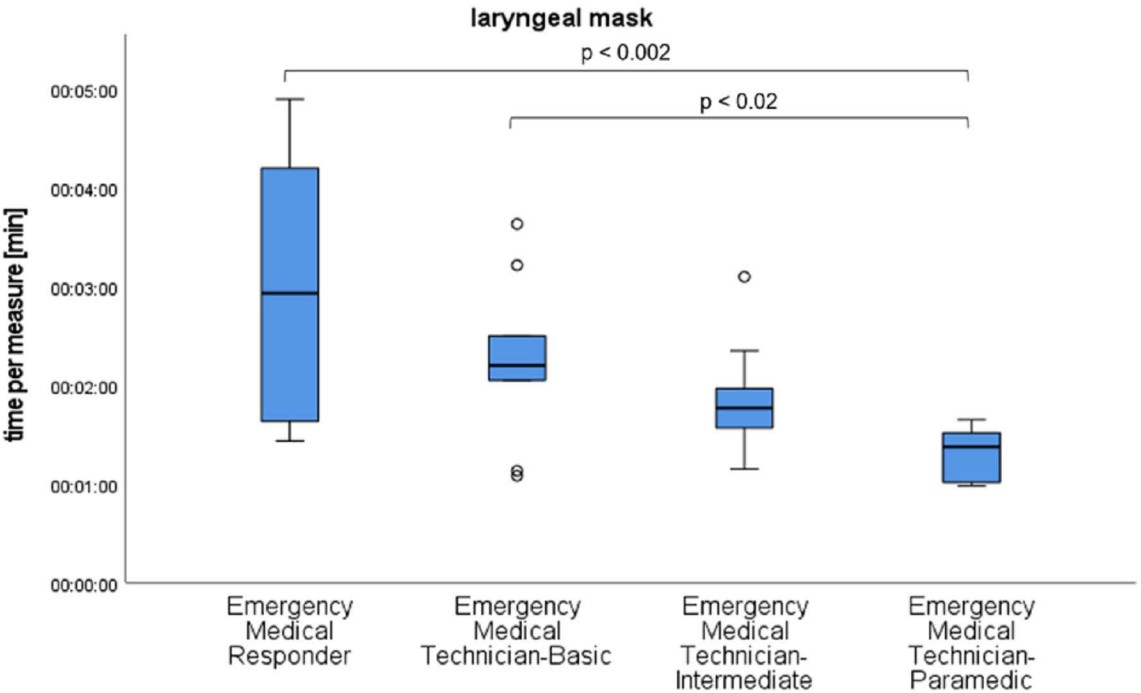

**Fig 3. Time results of the qualification subgroups for placement of a laryngeal mask with significant differences marked.**

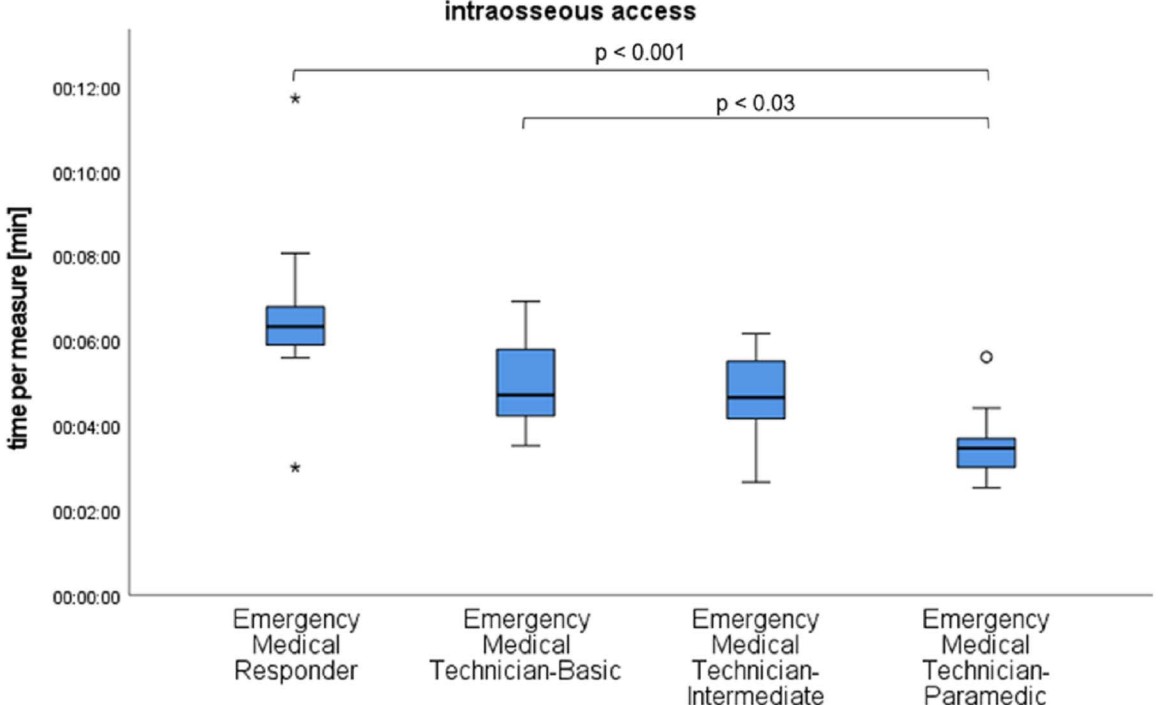

**Fig 4. Time results of the qualification subgroups for intraosseous access with significant differences marked.**

A total of four participants underdosed the medication (0.0 ml - 0.8 ml). The syringes were drawn up empty of air, which led to incorrect doses and was only noticed to the observer. This included two EMR participants and one each from the EMT-I and EMT-P groups. The highest failure rate was observed in establishing an I.O. access. Only 26 (66.7%) of the participants successfully completed the procedure under telemedical guidance. In addition to six EMR and four EMT-B participants, one participant from the EMT-I and three participants from the EMT-P also failed due to an incorrect position.

The Kruskal-Wallis test showed no significant differences in terms of success rates.

### Professional experience

Participants with more than ten years of experience, regardless of qualification, administered medication significantly faster than participants with five years of experience or less ($p < 0.046$). However, no influence of experience on time or success could be determined for the other interventions.

### Technical feasibility

The evaluation of the two different data glasses reveals that the HMT-1 had a better start with an average of 1.4 attempts compared to the M300, which required an average of 1.8 attempts to initiate. Furthermore, the HMT-1 showed more stability in its connection, with no interruptions recorded, whereas the M300 experienced an average of 0.6 interruptions. There is a statistically significant difference ($p = 0.02$) between the two data glasses regarding the number of interrupted teleconsultations during a simulation. In terms of camera position adjustments, the M300 performed better with an average of 3 position changes per participant compared to 4.2 changes with the HMT-1.

Fig 5 represents the results of the usability questionnaire of the data glasses. Regardless of the model, approximately half of the participants (57.5%) criticized the lack of secure attachment of the data glasses to the head. Concerning real-world use in civil protection, nearly half of the participants expressed fears of technical failures of the data glasses, and 25% expressed concerns about the loss of a stable infrastructure.

### Limitations

The study has shown that medical qualification is not a decisive indicator for the success of a medical intervention in CBRN hazardous situations delegated via telemedicine. However, it should be noted that the sample size was small with

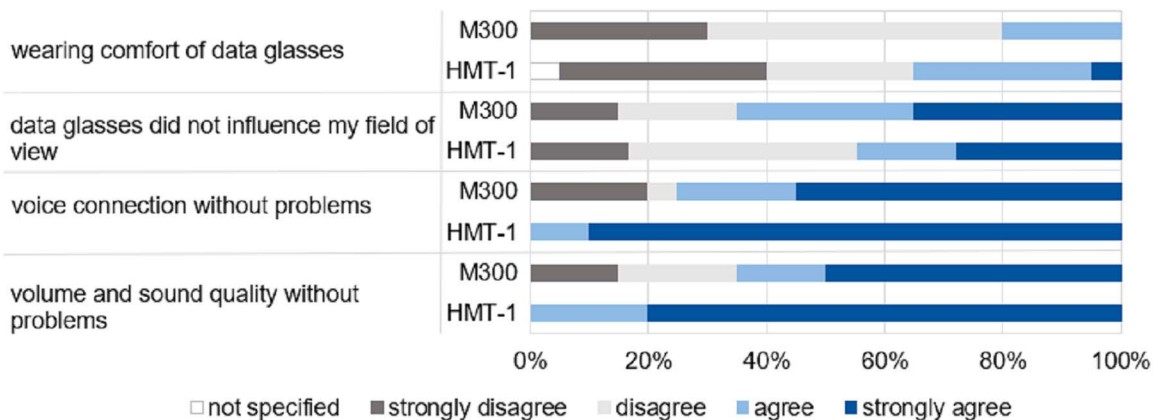

**Fig 5. Proband survey: Usability valuation of data glasses in CBRN protection suit (n=40).** The results of the Vuzix M300 (M300) and the Realwear HMT-1 (HMT-1) are listed.

only ten participants per group. Gathering additional participants was challenging, despite the extended study period. Further investigations with larger sample sizes should be conducted for more precise results.

Furthermore, it should be noted that conducting the study with a control group without telemedical support was not possible, as not all medical measures were included in the respective training catalog of the individual qualifications.

The simulation is of course limited in its informative value compared to real scenarios. In real CBRN situations, there is additional stress due to the imminent danger and the need for personal protection. The study to validate the potential of telemedical delegation of intraosseous access should also be carried out with a manikin that represents the anatomical conditions more realistically.

Due to an unforeseeable technical defect, the use of the different data glasses models was not evenly distributed across the groups with different qualifications, which is why an evaluation of the different models with an equally large sample size should be carried out in the future.

Since all participants were aware of being actively monitored and graded by an on-site observer and the telemedicine physician, the Hawthorne effect should be considered a possible bias [17]. However, since the study in this form was the first testing of telemedicine approaches in a CBRN scenario and especially while wearing a CBRN protection suit, a modified behaviour of the participants stimulated by the monitoring ist considered negligible.

Not all participants were provided with the appropriate protection suit, especially smaller protection suits and a range of different protective gloves were missing. Therefore, it can be assumed that the movement and thus the confidence in performing actions could be negatively affected by an ill-fitting suit. However, it is also to be expected that in a real scenario, limited consideration can be given to whether the sizes are a perfect fit, as only a limited number of protection suits items are available.

## Discussion

In this study, the potential of telemedically-guided interventions in CBRN missions was evaluated with differently qualified subjects. Although a comparison of all the qualifications analyzed revealed some significant differences in the time required, the success of the measures was independent of the qualification under telemedical guidance. However, both data glasses used are only suitable for use in the CBRN suit to a limited extent through a lack of user-friendliness and technical limitations due to the lack of setting options for the camera.

The different levels of education led us to expect that the responders would have different time requirements. More highly qualified responders require less time for medical measures, which can be emphasized by our study. However, it has also been shown that low-skilled responders can perform immediate life-saving measures in a reasonable time under telemedical guidance. Significant differences are only found between qualifications that differ more in training time, such as EMT-P to EMT-B and EMR, and EMT-I to EMR. Low-skilled responders often do not have knowledge or materials for interventions enabled by telemedicine. This is not part of their education curriculum. In order to successfully perform a medical intervention, it becomes an additional task for the telemedicine physician to name and describe the necessary materials and explain their functionality if they are not known to the responder.

The application of the tourniquet, similar to the nasal medication administration, was successfully performed by almost all participants. Compared to a previous study that provided just-in-time instructions on a 4x6-inch card as guidance for medical laypersons [18], the success rate was significantly increased through telemedical instructions. The time from injury to exsanguination in humans is not clearly defined and varies due to various factors [18]. However, Kragh et al. demonstrated in a previous study that the application of a tourniquet before the onset of hemorrhagic shock can ensure survival [19]. This measure is, without doubt, the most time-critical. Therefore, this life-saving immediate measure should be practiced by all emergency services. With regard to the errors made when applying the tourniquet, it should be noted that in some cases both the observers and the telemedicine physician assumed that the tourniquet had been applied correctly from their point of view, so that the simulation was continued according to the script. Only afterwards was it possible

to check whether the donning was correct. In a real case, the tourniquet would be applied until the bleeding was demonstrably stopped, which would lead to an even higher success rate.

The method of nasal drug application was deliberately chosen for the study due to its speed, simplicity and low risk. Failures were attributed to incorrect dosing, which was not visible to either the participant or the telemedicine physician due to limited visibility. The use of prefilled autoinjectors, as are already available and used on the market [20], could avoid the problem of incorrect dosing, even if this would eliminate the possibility of dose adjustment.

In a subgroup analysis, it was shown that professional experience had a significant influence on the time required to administer nasal medication. It can't be denied that it is easier to prepare and administer a medication using a syringe when you're experienced. However, no correlation was found between experience and time, number of attempts and success for the other tasks. Therefore, it cannot be assumed that experience is a criterion for determining which responders should use telemedicine.

It should be noted that the insertion of a laryngeal mask was performed correctly by all participants, regardless of their medical qualifications. Less qualified responders have usually only learnt manual ventilation with a bag valve mask during their training, the success rate of which depends on regular practice and experience. Stressful situations and the wearing of CBRN protection suits are factors that make ventilation with a bag valve mask more difficult for inexperienced responders. In contrast, ventilation via a correctly placed laryngeal mask proves to be a simple and safe method of providing adequate oxygenation. This is supported by a previous study involving laypersons, where successful laryngeal mask insertion and subsequent ventilation were achieved after a brief explanation [21]. The success rate in our study exceeded these results despite the presence of a CBRN protection suit, which has a negative impact on dexterity.

In all subgroups, there were difficulties in correctly performing the I.O. access. This was due not only to the inadequately realistic representation of anatomical conditions by the simulation mannequin but also to the lack of experience, particularly among the groups of EMR and EMT-B. Therefore, it was crucial for the participants to follow the instructions of the telemedicine physician and for the instructor to have a clear view of the patient's leg. Despite multiple corrections by the telemedicine physician, the error rate remained high. Incorrect positioning of the cannula can lead to compartment syndrome. Inappropriate application of force can result in a fracture. Incorrect handling can lead to infections or embolisms.

Considering the limited personnel resources in civil protection and the assumption that EMT-P personnel will be limited in civil protection scenarios, it is recommended to train other qualifications for the application of telemedicine. By creating a catalog of measures for each qualification, more invasive measures such as the I.O. access can be reserved for EMT-I and EMT-P. Education programs, simulation exercises, and the development of Standard Operation Procedures (SOPs) could help educate more telemedicine physicians and responders for telemedicine deployment in CBRN situations, focused on the EMT-I as the most suitable responders. Previous studies suggest a need in this regard [22].

The use of data glasses has revealed numerous issues that make its usage in CBRN protection suits not advisable. It is evident that both models are not compatible with CBRN protection suit, as neither repositioning after displacement nor adjusting the camera position after donning the suit was possible. A misfocussing of the data glasses could also not be rectified. The ability for the telemedicine physician to control the camera is lacking, as well as the incorporation of a zoom function to allow for more detailed monitoring of the measures. The challenges in starting the models make their use unreliable tools for real-world deployment. However, only the data glasses made some medical procedures possible in this study. Their use should be considered a success if some medical measures or therapies would otherwise fail to materialize.

Respondents consistently express concerns about unstable internet connections in disaster situations, as in the study by Zhang et al. [23] Cellular networks or the use of satellite communication could be helpful [24].

One possible solution could be the use of a smartphone as a technical alternative to data glasses in combination with the use of an external camera and a headset. In addition to their user-friendly nature, smartphones highly score thanks to their voice control and hands-free function as well as their ability to connect to external technical devices. It would be

conceivable to control them using sufficiently large control buttons that are for example built into a wristband and can be worn outside the CBRN protection suit.

The data glasses with display were not convincing in the study, as they led to a relevant impairment of the user's field of vision. The use of an external camera with an adjustable strap is therefore recommended. This allows the camera to be positioned on the forehead and thus outside the user's field of vision. Alternatively, integration into the visor of the CBRN protection suit could also be considered. Irrespective of the technical limitations, the evaluation of the medical measures shows that almost all of them have already been successfully guided with telemedicine using data glasses. The data glasses have thus already laid a positive foundation for a technology that enables more comprehensive medical care than current regulations would allow in Germany, for example, even in the event of a shortage of physicians.

## Conclusion

To summarize, it can be said that delegable medical measures can also be guided by telemedicine in CBRN protection. Despite some significant time differences between lower and higher qualified responders, the success rate shows that, in principle, all civil protection responders can be instructed via telemedicine.

We recommend the telemedical training of the EMT-I, as no significant difference was found in comparison to the EMT-P. The EMT-I is the highest non-medical qualification in German civil protection. They have previous experience from their education, which could be expanded in supplementary training through measures for telemedicine.

However, due to technical limitations of the data glasses such as a lack of autofocus or the inability to subsequently correct the position in the CBRN protection suit, the data glasses prove to be unsuitable in CBRN scenarios. Alternative technical solutions must be developed here.

## Author contributions

**Conceptualization:** Sarah Bovenkerk, Michael Czaplik, Andreas Follmann.

**Data curation:** Sarah Bovenkerk, Andreas Follmann.

**Formal analysis:** Sarah Bovenkerk.

**Funding acquisition:** Rolf Rossaint, Michael Czaplik, Andreas Follmann.

**Investigation:** Sarah Bovenkerk, Anna Mueller, Andreas Follmann.

**Methodology:** Sarah Bovenkerk, Anna Mueller, Michael Czaplik, Andreas Follmann.

**Project administration:** Anna Mueller, Rolf Rossaint, Andreas Follmann.

**Resources:** Rolf Rossaint, Michael Czaplik, Andreas Follmann.

**Software:** Michael Czaplik.

**Supervision:** Rolf Rossaint, Michael Czaplik, Andreas Follmann.

**Validation:** Anna Mueller, Rolf Rossaint, Andreas Follmann.

**Visualization:** Sarah Bovenkerk, Andreas Follmann.

**Writing – original draft:** Sarah Bovenkerk.

**Writing – review & editing:** Anna Mueller, Rolf Rossaint, Michael Czaplik, Andreas Follmann.

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
