## [Decision Letter · Decision Letter 0]

18 Mar 2025

PONE-D-24-49405Telemedicine via data glasses in CBRN protection suit – Evaluation of medical qualification and technical feasibilityPLOS ONE

Dear Dr. Bovenkerk

Thank you for submitting your manuscript to PLOS ONE. After careful consideration, we feel that it has merit but does not fully meet PLOS ONE’s publication criteria as it currently stands. Therefore, we invite you to submit a revised version of the manuscript that addresses the points raised during the review process.

Please submit your revised manuscript by May 02 2025 11:59PM** ,** If you will need more time than this to complete your revisions, please reply to this message or contact the journal office at plosone@plos.org . Please include the following items when submitting your revised manuscript:

We look forward to receiving your revised manuscript.

Kind regards,

Thiago Gonçalves dos Santos Martins

Academic Editor

PLOS ONE

Journal Requirements:

“Funded by Federal Office of Civil Protection and Disaster Assistance during the TeleSAN project (FKZ: 41201/45).”

“MC (CEO, co-founder) and RR (co-founder) are with Docs in Clouds TeleCare GmbH, a company developing telemedicine software. The other authors declare no conflicts of interest.”

Additional Editor Comments :

Dear Sarah,

We are pleased to inform you that the review process for your manuscript, Telemedicine via data glasses in CBRN protection suit – Evaluation of medical qualification and technical feasibility

, has been completed. The reviewers have provided their comments and suggestions, which are now available for your consideration.

We kindly request that you review their feedback and submit your responses along with the revised version of your manuscript. Addressing these comments thoroughly will help strengthen your work and enhance its overall quality.

Please submit your revisions by [Deadline Date]. If you require additional time or have any questions regarding the reviewers’ comments, do not hesitate to reach out.

We appreciate your contributions and look forward to receiving your revised manuscript soon.

Reviewer 1:

Title: Telemedicine via data glasses in CBRN protection suit – Evaluation of medical qualification and technical feasibility

Overall, this was a well written and very interesting study on possible applications of data glasses in an area of need.

Syntax/Grammar

-Recommend rephrasing lines 32 and 33 of the Results section of the abstract to improve clarity

-Recommend rephrasing line 239 of page 11 (section 3.1) to improve clarity

Abstract

-Please spell out CBRN in the abstract for those who may not read the full manuscript

-After stating “data glasses” in the abstract and title, consider adding in parentheses “smart glasses” which may be better understood in some countries. For example, data glasses (smart glasses)

-You may also consider adding additional terms for protection suit (more commonly referred to as a hazmat suit in the US).

Introduction

-On page 4 line 73 could you please list a few examples of medications (such as pain meds or meds needed for blood pressure management or all meds?)

-The last two sentences of the introduction on page 5 lines 100-102 would better be moved to the Results or Conclusion sections

-It may be useful to include information (if available) about what currently happens in emergency situations when a physician is not present on site. For example, do patients simply not undergo any needed interventions and not receive any medications until appropriate staff is present? Are other forms of telemedicine such as telephones, smart phones or real-time messaging used that would fulfill the legal requirement?

Methods

-In section 2.1, it would be helpful if clarification on the first questionnaire could be given. Specifically, is this the same as the questionnaire given after the simulation and if not, what types of questions were on the questionnaire

-Please specify in section 2.4 if both types of data glasses also give real-time audio to the telemedicine physician

Results

-Good

Limitations

-As participants were actively monitored and graded by an on-site observer and the telemedicine provider, it may be worth mentioning the Hawthorne effect as a possible bias

Discussion

-While it is notable from the technical difficulties that the data glasses were not a viable solution, it is worth noting that they allowed these medical procedures at all. If the alternative is patients not receiving any urgent/essential medical care then it could be said that the telemedicine solution is a success and that in addition to smart phones, any other telemedicine platform that fulfills the legal requirement whether it be radio (which may be more reliable in disaster situations) or synchronous text messaging would be far superior to the current model

Tables/Figures

-Good

Reviewer 2:

This is an original research. Methods were described in sufficient detail The conclusions are supported by the methods applied. Statistical analysis are coherent with the objectives/study design. The Authors made available data used in the study.

Reviewers' comments:

Reviewer's Responses to Questions

**Comments to the Author**

1. Is the manuscript technically sound, and do the data support the conclusions?

Reviewer #1: Yes

Reviewer #2: Yes

2. Has the statistical analysis been performed appropriately and rigorously? 

Reviewer #1: I Don't Know

Reviewer #2: Yes

3. Have the authors made all data underlying the findings in their manuscript fully available?

Reviewer #1: Yes

Reviewer #2: Yes

4. Is the manuscript presented in an intelligible fashion and written in standard English?

Reviewer #1: Yes

Reviewer #2: Yes

5. Review Comments to the Author

Reviewer #1: Title: Telemedicine via data glasses in CBRN protection suit – Evaluation of medical qualification and technical feasibility

Overall, this was a well written and very interesting study on possible applications of data glasses in an area of need.

Syntax/Grammar

-Recommend rephrasing lines 32 and 33 of the Results section of the abstract to improve clarity

-Recommend rephrasing line 239 of page 11 (section 3.1) to improve clarity

Abstract

-Please spell out CBRN in the abstract for those who may not read the full manuscript

-After stating “data glasses” in the abstract and title, consider adding in parentheses “smart glasses” which may be better understood in some countries. For example, data glasses (smart glasses)

-You may also consider adding additional terms for protection suit (more commonly referred to as a hazmat suit in the US).

Introduction

-On page 4 line 73 could you please list a few examples of medications (such as pain meds or meds needed for blood pressure management or all meds?)

-The last two sentences of the introduction on page 5 lines 100-102 would better be moved to the Results or Conclusion sections

-It may be useful to include information (if available) about what currently happens in emergency situations when a physician is not present on site. For example, do patients simply not undergo any needed interventions and not receive any medications until appropriate staff is present? Are other forms of telemedicine such as telephones, smart phones or real-time messaging used that would fulfill the legal requirement?

Methods

-In section 2.1, it would be helpful if clarification on the first questionnaire could be given. Specifically, is this the same as the questionnaire given after the simulation and if not, what types of questions were on the questionnaire

-Please specify in section 2.4 if both types of data glasses also give real-time audio to the telemedicine physician

Results

-Good

Limitations

-As participants were actively monitored and graded by an on-site observer and the telemedicine provider, it may be worth mentioning the Hawthorne effect as a possible bias

Discussion

-While it is notable from the technical difficulties that the data glasses were not a viable solution, it is worth noting that they allowed these medical procedures at all. If the alternative is patients not receiving any urgent/essential medical care then it could be said that the telemedicine solution is a success and that in addition to smart phones, any other telemedicine platform that fulfills the legal requirement whether it be radio (which may be more reliable in disaster situations) or synchronous text messaging would be far superior to the current model

Tables/Figures

-Good

Reviewer #2: This is an original research. Methods were described in sufficient detail The conclusions are supported by the methods applied. Statistical analysis are coherent with the objectives/study design. The Authors made available data used in the study.

---

## [Author Response · Author response to Decision Letter 1]

4 Apr 2025

Response to Reviewers

Dear Sir or Madam,

thank you very much for taking the time to read my manuscript and for our comments and improvements. We have edited the manuscript accordingly and made the following changes:

Financial disclosure

The funding code of the study was incorrect. We request the following change in the submission portal. We have also added that the sponsor had no influence on the study with your proposal:

“Funded by the Federal Office of Civil Protection and Disaster Assistance as part of the TeleSAN project (FKZ: 41201/425). The sponsor had no influence on the study design, the data collection and analysis, the decision to publish or the preparation of the manuscript.“

Compensating interests

We request the following change in the submission portal:

“MC (CEO, co-founder) and RR (co-founder) are with Docs in Clouds TeleCare GmbH, a company that develops telemedicine software. This does not change our adherence to PLOS ONE policies on sharing data and materials. The other authors declare no conflicts of interest.”

Formatting

The formatting of the manuscript has been changed according to the PLOS ONE style guide.

Abstract

The word CBRN was written out to make it easier to understand for readers who only read the abstract.

The terms “hazmat suits” and “smart glasses” have been added to make it easier to understand in international context.

According to reviewer 2, we have changed two sentences in the abstract for better understanding.

Introduction

Drug therapies in emergency situations are severely restricted by non-medical emergency personnel. Two substances were listed as examples.

In the last section of the introduction, the current situation regarding the administration of medication by EMT-P and other non-medical emergency personnel in Germany was explained.

Reviewer 1 has noted that the last sentences of the introduction should be better moved to the conclusion. Due to the submissions guidelines of PLOS Onne, who would like a preview of the results in the introduction, it was decided not to postpone the two sentences.

Methods

Thank you for your comments on the methodology section. A detailed description of the questionnaires has been included to improve the presentation of the study design. As described, the study participants were selected in equal proportions according to their qualifications.

In addition, the information that both data glasses transmitted the sound in real time was added to the “Telemedicine” section.

The patient was monitored on site with the help of a surveillance monitor. The practical performance of vital sign measurements was not part of the study investigation, as all participants were appropriately trained.

Limitations

Due to the study design, a Hawthorne effect could not be ruled out with certainty. This aspect was included in the limitations. We have added the reference [17] for this purpose.

Discussion

According to Figure 5, the HMT-1 does not show poor audio quality, so it is not necessary to add a headset at this point. A headset could be used as a supplement for other glasses; this has been added accordingly in the text referring to Reviewer 2.

Reviewer 2 commented on whether additional medical equipment would be useful for telemedicine. As medical devices have to be cleaned or disposed of at great expense after use in contaminated areas, additional material is avoided wherever possible. In addition, a conceptual attempt is made to decontaminate the patient as quickly as possible. We have therefore dispensed with further equipment in the study and have not included this point in the paper. If you have any additional comments or requests for changes, please let us know.

The following point was included in the discussion: Despite the technical difficulties, it should not be ignored that the smart glasses have made life-saving measures possible in the first place, which would not be possible without telemedicine.

References

The references has been checked. Since reference [4] is no longer available, it was replaced by a reference with the same content.

A corresponding reference to the Hawthorne effect was added.

Tracking changes of the references is unfortunately not possible, so the changes in the references have been highlighted accordingly.

---

## [Editor Report · Decision Letter 1]

27 Apr 2025

Telemedicine via data glasses in CBRN protection suit – Evaluation of medical qualification and technical feasibility

PONE-D-24-49405R1

Dear Dr. Bovenkerk,

We’re pleased to inform you that your manuscript has been judged scientifically suitable for publication and will be formally accepted for publication once it meets all outstanding technical requirements.

Kind regards,

Thiago Gonçalves dos Santos Martins

Academic Editor

PLOS ONE
---

## [Editor Report · Acceptance letter]

PONE-D-24-49405R1

PLOS ONE

Dear Dr. Bovenkerk,

I'm pleased to inform you that your manuscript has been deemed suitable for publication in PLOS ONE. Congratulations! Your manuscript is now being handed over to our production team.

Kind regards,

on behalf of

Dr. Thiago Gonçalves dos Santos Martins

Academic Editor

PLOS ONE